# Role of Melatonin in Viral, Bacterial and Parasitic Infections

**DOI:** 10.3390/biom14030356

**Published:** 2024-03-16

**Authors:** Georges J. M. Maestroni

**Affiliations:** Center of Research in Medical Pharmacology, University of Insubria, 21100 Varese, Italy; georges.maestroni@tim.it

**Keywords:** melatonin, circadian rhythm, immunomodulation, anti-inflammatory effect, cytokine storm, infection

## Abstract

In all mammals, the circulating pool of MLTs is synthesized in the pineal gland during the night’s darkness hours. Its main function is synchronizing the organism in the photoperiod. In contrast, extra-pineal MLT is synthesized in peripheral organs, does not follow any circadian rhythm or circulate, and plays a detoxifying and cytoprotective role. Circulating MLT may stimulate both innate and acquired immune responses through its circadian action and by activating high-affinity receptors on immunocompetent cells. Extra-pineal MLT may have antioxidant and anti-inflammatory effects that dampen the innate immune response. These two seemingly divergent roles may be considered to be two sides of the same coin. In fact, the integration of both circulating and extra-pineal MLT functions might generate a balanced and effective immune response against microbial pathogens. The studies described in this review investigated the effects of exogenous MLT in various models of infectious diseases using extremely different doses and treatment schedules. None of them evaluated the possibility of integrating the non-circadian anti-inflammatory effect with the circadian immunoenhancing action of MLT. As a consequence, in spite of the fact that most studies agree that MLT has a beneficial effect against infections, it seems difficult to draw any definite conclusion about its possible therapeutic use.

## 1. Introduction

The current view of the origin of melatonin (MLT), chemically defined as N-acetyl-5-methoxytriptamine, suggests that MLT appeared on earth about 2.5 billion years ago. Indeed, it is proposed that at that time, anaerobic bacteria developed the ability to synthesize MLT as an adaptative response to increasing concentrations of oxygen in the atmosphere. Thus, the very first function of MLT is considered to have been counteracting oxygen toxicity. These bacteria were then eventually phagocytized by eukaryotes where, according to the endosymbiotic theory, they established a symbiotic association and evolved in mitochondria or chloroplasts, retaining the ability to synthesize MLT [1]. This could explain why MLT may be synthesized in many, if not in all, extra-pineal organs. In mammals, MLT has been identified in skin, gastrointestinal tract, liver, kidney, the immune system, bone marrow, the testes, skeletal muscles and all body fluids [2]. Related to the topic of this review, it is worth mentioning that cutaneous MLT seems to be involved in defense against shin infections [3,4]. In general, the concentration of extra-pineal MLT is reported to be several orders of magnitude higher than that of the circulating pool derived from the pineal gland. Nevertheless, it has recently been found that the gastrointestinal tract is not a major source of extra-pineal MLT, as was formerly believed [5]. It is, therefore, possible that some of the early studies in this field overestimated the MLT content because of methodological flaws. 

Beyond being able to act as an antioxidant [6], during evolution, MLT acquired many other functions and became a pleiotropic molecule. In vertebrates, MLT has acquired the basic function of synchronizing the organism’s physiology with the 24-h environmental cycle (circadian rhythm) caused by the daily rotation of our planet. This vital effect is carried out by the circadian oscillation of MLT synthesis in the pineal gland accompanied by its immediate release into the blood circulation [7]. In contrast, extra-pineal MLT does not show any rhythmicity, and it is not secreted into the blood in significant amounts [2]. The environmental cue driving the MLT circadian rhythm is the light/dark cycle of the day. In particular, light is sensed in retinal ganglion cells through a photopigment called melanopsin that is activated by photons of 460–480 nm in wavelength (blue light). The ensuing nervous signal travels in the retino-hypothalamic tract and entrains the suprachiasmatic nucleus (SCN) of the hypothalamus, i.e., the central biological clock of the organism. In turn, the SCN activates a nervous pathway involving the paraventricular nucleus (PVN) of the hypothalamus, the intermediolateral cell column and the superior cervical ganglia (SCG) regulating MLT synthesis in the pineal gland [7]. Remarkably, blue light inhibits MLT synthesis; hence, MLT is synthesized and released during the darkness part of the daily photoperiod in all vertebrates [1,7], and this synchronization occurs in the organism in this part of the photoperiod by activating specific MLT receptors in both the SCN and peripheral biological clocks [1,7]. Basic biological rhythms such as the oscillations of the autonomic nervous system activity and the hypothalamic pituitary adrenal (HPA) axis are also driven by the SCN upon the synchronizing action of MLT [8].

In vertebrates, MLT synthesis is attained by two enzymatic steps: the first is accomplished by serotonin N-acetyl transferases (SNATs) that lead to the formation of N-acetyl-serotonin (NAS), and the second transforms NAS in MLT using the enzyme hydroxyindole-O-methyl transferase (HIOMT) [1].

## 2. MLT Receptors

Two high-affinity membrane receptors, MT1 and MT2, have been cloned from mammalian tissues, including human tissues. They share specific amino-acid sequences, indicating that they belong to a specific receptor subfamily [9,10]. Both are G-protein-coupled receptors (GPCR) activating similar signaling pathways, which result in the inhibition of adenyl cyclase and activation of phospholipase C influencing gene expression. However, GPCRs may also physically associate with intracellular proteins other than G proteins, and this may specifically allow the targeting of a cellular compartment with different outcomes. Furthermore, MT1 and MT2 may interact by forming heterodimers [11]. Another member of the MT1 and MT2 subfamilies is the melatonin-related receptor, also known as GPR50. This receptor is an orphan receptor because it does not bind MLT and its endogenous ligand is unknown. However, GPR50 can heterodimerize with both MT1 and MT2. Dimerization with MT1, but not with MT2, blocks MLT signaling [12]. A third low-affinity MLT binding site situated in the cytosol is the enzyme quinone reductase (QR2), whose activity may be related to the antioxidant and protective effects of MLT [11,13]. Finally, MLT can bind with the high-affinity calcium-binding proteins calmodulin and calretinin, thereby affecting the cell cycle [14], a phenomenon that has been related to the oncostatic action of MLT [15]. 

Due to its amphipathic nature, MLT can easily reach the cell nucleus and, according to many reports, bind members of the retinoid acid-related nuclear receptor (ROR) family [16]. ROR is strictly associated with the pleiotropic effect of MLT in various physiological and pathological processes, including circadian rhythms, immunity, inflammation, oxidative stress, and oncogenesis [16]. Studies have recently contested the conceptualization of RORs as MLT receptors [16]. However, the fact that both ROR and MLT are dependent on similar signaling pathways and have identical functions suggests that MLT can modulate ROR expression and function via indirect mechanisms involving MT1 and MT2 receptors or other mediators [16].

## 3. MLT as an Antioxidant

Oxidative stress can be caused by excessive production of reactive oxygen species (ROS) or reduced activity of the antioxidant system. Oxidative stress is well known to increase inflammation and contribute to a variety of pathological conditions, including cancer, cardiovascular diseases, neurodegenerative diseases, lung diseases, renal diseases and aging. MLT is considered to be a major player in the antioxidant machinery because of its direct scavenging of ROS and its stimulation of antioxidant enzymes and suppression of pro-oxidant enzymes [17]. Its direct effect as a scavenger of free radicals has been clearly demonstrated in cell cultures, where MLT and its metabolites usually added at pharmacological concentrations may act by a variety of mechanisms, including electron transfer, hydrogen transfer and metal chelation [18]. However, it has been recently pointed out that in living organisms, the amount of substances that may react with MLT and its metabolites largely exceeds their concentration, even considering that the concentration of extra-pineal MLT is usually several orders of magnitude higher than that of plasma MLT [19]. This simple stoichiometric consideration casts some doubt on the conceptualization of MLT as an all-purpose in vivo scavenger of free oxygen or nitrogen radicals. However, a different interpretation suggests that MLT and its metabolites may act as stabilizers of the redox state of mitochondria when energy is produced via mitochondrial oxidative phosphorylation [18]. On the contrary, it is well recognized that the activation of MT1 and MT2 receptors enhances the expression of antioxidant enzymes such as superoxide dismutase, catalase, glutathione peroxidase and glutathione reductase [17,20,21]. In addition, the MLT binding to QR2 inhibits its enzymatic activity, reducing the generation of ROS [13].

## 4. Melatonin and the Immune System

Few reports associated the pineal gland with the immune system [22,23] before 1986, when it was shown, for the first time, that MLT could increase antibody production in mice and counteract the immunosuppressive effect of corticosterone and/or restrain stress via an opiatergic mechanism [24,25,26]. These findings were reproduced in different experimental models [27,28], and, in general, the immunoregulatory properties of MLT have been further extended in a variety of animal models and in humans [29,30,31,32,33,34,35,36,37]. Nevertheless, the overall picture of the immunological action of MLT is quite confused. Several reports indicate that MLT is a powerful in vivo immunoenhancing factor, suggesting its use as a therapeutic agent whenever it is needed to boost humoral and/or cellular immune responses, while others endow MLT with an anti-inflammatory effect [29,30,31,32,33,34,35,36,37]. This apparent contradiction might be due to the wide range of concentrations and dosages used, possibly linked to the pleiotropic nature of the molecule or, more probably, to the fact that the immunological consequences of the circadian action exerted by MLT via its specific receptors were seldom discriminated from the other non-circadian effects [38].

It is well known that the immune system is under a circadian control exerted by the SCN, which drives the activities of the sympathetic nervous system (SNS) and the hypothalamo-pituitary adrenal (HPA) axis. Both the SNS and HPA axis convey circadian information to peripheral organs by regulating clock gene expression [39,40] and basic immunological functions, such as the blood circulation of immunocompetent cells, their infiltration into peripheral organs and the circadian oscillation of their specific functions [41]. In general, circulating cells peak in the blood during the resting phase of the photoperiod, while their migration into peripheral tissues occurs during the active phase. These phenomena are essential to ensure tissue homeostasis and activate the appropriate immune response in case of infection. For example, it has been reported that lymphocyte migration into lymph nodes peaked in the phase with dendritic cells (DC) at the beginning of the active phase to optimize antigen presentation and the ensuing adaptive immune response [42]. In this context, MLT and its receptors play a fundamental role due to their ability to synchronize the circadian output of the SCN and/or drive circadian rhythms directly in other brain structures [43]. Thus, the immunoenhancing action of MLT is linked to its circadian properties, including the receptor-mediated modulation of cytokine production, cell migration and antigen presentation to immunocompetent cells [44,45]. Last but not least, MLT may suppress the nuclear translocation of glucocorticoid receptors [46] and, hence, modulate their effect on immunity [47].

In conclusion, the available evidence, including exogenous administration and studies in pinealectomized animals [45], suggests that the immunoenhancing action of MLT is exerted at physiological or supraphysiological concentrations via the activation of its specific receptors (Figure 1). 

Blue light constitutes the major environmental cue regulating MLT synthesis in the pineal gland by inducing the nocturnal activation of preganglionic sympathetic neurons. MLT produced during the darkness hours is released into the blood and synchronizes the central biological clock of the organism, i.e., the SCN, with the photoperiod. In turn, the SCN drives the circadian oscillation of the sympathetic nervous system, which may influence immunocompetent cells via the adrenergic activation of α-and β-adrenoceptors. The SCN rhythm also induces the paraventricular nucleus to release corticotropin-releasing hormone, which, in turn, stimulates adrenocorticotropic hormone production by the anterior hypophysis. This final step stimulates the production of glucocorticoids(GCs) by the adrenal cortex. All together, these mechanisms influence the circulation, migration and functions of immunocompetent cells. In addition, a robust circadian machinery is associated with regular sleep, with positive effect on immunity. Last but not least, circulating MLT may act directly on MT1 and MT2 expressed on immunocompetent cells, as well as on calmodulin (CAM), and modulate the expression of cell adhesion molecules, cytokine production and cell proliferation. The circadian administration of exogenous MLT using amounts that do not oversaturate its receptors may reinforce both the indirect and direct immunoenhancing effects of MLT.

On the contrary, non-circadian effects that comprise anti-inflammatory and mitochondria-related effects are exerted by MLT at concentrations in the same order of magnitude as extra-pineal MLT that physiologically does not contribute to the circulating pool. As such concentrations would oversaturate any receptor, the aforementioned effects are obviously receptor-independent [35,48]. 

Another important consideration concerning the immunological action of MLT relates to the widely used distinction between pro-inflammatory and anti-inflammatory effects, with the former being used synonymously with “immunostimulating”. This effect involves the stimulation of cytokines such as IL-1β, IL-2, IL-6, IL-8, TNF-α, IFNγ and IL-17A and/or the upregulation of cyclooxygenase-2 (COX-2) and inducible NO synthase. Other effects are exerted on hematopoiesis by stimulating GM-CSF and the differentiation of Th cells and NK cells [46]. Yet, an excessive inflammatory response may have a paradoxical effect on immunity, leading to the activation of the coagulation system, organ failure and immunodeficiency by inducing T-cell apoptosis [49]. Several studies bestow MLT with the capacity of exerting opposite actions based on the ongoing biological process [50]. With regard to the immune system, it has been proposed that the contrasting actions of MLT represent a system to guarantee the appropriate immune response according to the pathological situation [33,49]. The anti-inflammatory action of MLT includes several mechanisms in part related to its antioxidant properties. MLT may inhibit NF-kB activation, upregulate Nrf2 and inhibit TLR4 signaling. Some of these MLT effects seem to be related to the activation of sirtuin 1 (SIRT1) [48].

Bidirectional communication between the pineal gland and the immune system has been proposed as a mechanism for integrating the immunological functions of both pineal and extra-pineal MLT. This mechanism, termed the immune–pineal axis, is based on the transfer of MLT production from the pineal gland to local immunocompetent cells at the site of infection or tissue damage to control the inflammatory response and then, after resolution, back to the pineal gland [51]. The transient inhibition of pineal MLT production and the induction of its synthesis in macrophage/microglia seems to depend on NFkB activation by circulating cytokines and/or pathogen-associated molecular patters (PAMPs). Then, the inhibition of NFkB activity by extrapineal MLT and activation of the HPA axis associated with the inflammatory response restore the circadian release of pineal MLT, which is essential for optimizing the acquired immune response and maintaining the immune homeostasis [51]. Thus, we could infer that the anti-inflammatory and pro-inflammatory effects of MLT are two sides of the same coin, aimed at ensuring a successful immune response against the invading pathogen by balancing the innate with the acquired immune response.

Besides immunity, MLT can influence hematopoiesis, i.e., the process responsible for the daily production of erythrocytes and immunocompetent cells. MLT was shown to rescue hematopoiesis in mice against the toxicity of anti-cancer drugs. This effect was apparently due to Th-cell-derived opioid cytokines binding to the k-opioid receptor on GM-CSF-activated bone marrow (BM) stromal cells, possibly resulting in IL-1 production [52]. The ability of MLT to counteract myelosuppression due to the toxic action of anti-cancer drugs was then amply confirmed in patients [53,54]. In BM, hematopoietic stem and progenitors cells’ (HSPCs) circadian mobilization and circulation are essential for replenishing the blood with immunocompetent cells and ensuring the immune system’s homeostasis. In this process, BM-derived MLT plays a major role by inducing HSPC quiescence and retention [55].

Finally, MLT may influence the immune system via its well-known effect on sleep. Sleep and immunity are tightly linked. Regular sleep is crucial for the immune system, and immune-derived factors are needed for regular sleep. Thus, the effect of MLT on inflammatory cytokines might be linked to its sleep-facilitating action, which, in turn, contributes to maintaining a healthy immune system [56].

## 5. MLT and Viral Infections

The first evidence of an antiviral activity of MLT was shown against encephalomyocarditis virus (ECMV), a highly pathogenic virus that produces encephalitis and myocarditis in rodents. Exogenous MLT could prevent the paralysis and deaths of mice infected with EMCV [57]. Other encephalitogenic viruses proved to be affected by MLT. Normal mice were infected with the Semliki Forest virus (SFV), and stressed mice were injected with the attenuated non-invasive West Nile virus (WNV). SFV can produce viral encephalitis in normal mice, while the attenuated form of WNV can do it only in immunosuppressed mice. In both models, the administration of MLT significantly postponed the onset of the disease and reduced mortality [58]. A similar effect was then reported in mice infected with Venezuelan equine encephalomyelitis virus (VEEV) [59]. The protective effect of MLT in this model was shown to depend on increased IL-1β production, as it was abolished by IL-1β neutralization [60]. An inverse correlation between MLT and IL-12 plasma levels and disease progression has been described in HIV-1-infected individuals, suggesting a direct relationship between MLT and Th1 cell function [61]. MLT has also proven to be effective against respiratory syncytial virus (RSV). The in vitro infection of macrophages with RSV-activated TLR3 and NFkB and the subsequent inflammatory response was also identified. In this model, MLT was able to inhibit the response by suppressing NFkB activation [62]. This effect was reproduced in mice infected with RSV, where MLT could inhibit lung oxidative stress [63]. The anti-inflammatory and regenerative effects of MLT were also evident in rabbit with fulminant hepatitis of viral origin [64]. In the same model, another study showed that MLT could inhibit mitophagy and the innate immune response while restoring the circadian dysregulation induced by the virus, recommending the use of MLT as a therapeutic option in human fulminant hepatic failure [65]. In an in vitro model of Hemorrhagic Shock Syndrome caused by the EBOLA virus, MLT was highlighted as a promising therapeutic agent because of its ability to neutralize endothelial cell disruption [66].

With the advent of the COVID-19 pandemic, an impressive number of studies have tested MLT for possible therapeutic and prophylactic effects. A PubMed search conducted with the terms MLT and COVID-19 retrieved 138 publications, including many clinical randomized studies. However, in spite of this outsized number of publications, it is difficult to draw any definite conclusion about the therapeutic efficacy of MLT in COVID-19 patients. In fact, there are reports showing the positive therapeutic effect of exogenous MLT [67,68,69,70,71,72], while others deny any effect [73,74,75]. Amid the beneficial effects exerted by MLT against COVID-19, we can find the prevention of complications and reduction in mortality in severely ill patients [69,72], improvement in respiratory symptoms via the reduction in the lungs’ involvement [71] and reduced requirement for invasive mechanical ventilation, as well as overall improvement in clinical status [72]. On the other hand, a randomized retrospective study negates any effect of MLT on survival of COVID-19 patients [73], and another contemporaneous randomized clinical trial reached the same conclusion [74]. Perhaps this drastic discrepancy is due to the wide array of doses and treatment schedules used in these studies which continue to perpetuate misconceptions about the real therapeutic properties of MLT. For example, MLT has been administered once per day in the evening at a 10 mg dose for 14 days [69] or twice per day without mentioning the timetable at a dose-pro-dose of 3 mg [69] or 5 mg [70] in these studies. In the rationale of the studies, no authors considered a possible distinction between the circadian and non-circadian effects of MLT that could be related to its conceivable therapeutic effect against SARS-CoV-2. Most studies just mentioned, in a general fashion, the immunomodulatory and anti-inflammatory effects of MLT. Moreover, some new and peculiar mechanisms of action have been highlighted to explain the observed effects of MLT. Thus, the influence of MLT on the pathogenic enzyme p21-activated kinase 1, whose activation is involved in a variety of pathological conditions including viral infections [67], cluster differentiation 147 [68], viral phase separation and epitranscriptomics [70] and the coagulation system [69], has been reported.

The emergency linked to the COVID-19 pandemic has somewhat boosted interest in the putative antiviral potential of MLT, generating studies about its effects on influenza infections. Even in this case, MLT has been administered at very high doses and, in some cases, with treatment schedules ignoring completely its circadian nature. A report claims that MLT ameliorates influenza A H1N1 infection in mice by virtue of its ability to inhibit pro-inflammatory cytokines while enhancing the anti-inflammatory cytokine IL-10. MLT was administered subcutaneously either 6 h before infection and/or 2, 4 and 6 days post-infection at a concentration of 200 mg/kg b.w. without specifying any timetable [76]. In another study of mice infected with influenza A H3N2, MLT was administered intraperitoneally at 30 mg/kg b.w. for 7 days in the evening. In this case, MLT was proved to reduce pulmonary damage, leukocyte infiltration and edema and switch the polarization of alveolar macrophages from the M1 to the M2 phenotype [77]. A third study provided the interesting observation that MLT-deficient mice show a significantly higher mortality rate in comparison to their wild-type counterpart after infection with influenza A H1N1 virus. In other experiments, BALB/c mice were pretreated for 3 days via the intranasal administration of MLT (3, 10 and 30 mg/kg b.w.) before virus inoculation. The MLT-treated animals were apparently significantly protected from the virus by the suppression of mast cell activation and inhibition of cytokine storm [78].

Again, we are faced with results that are difficult to integrate into a clear understanding of MLT’s action. In particular, it seems rather problematic to combine the interesting observation of an augmented vulnerability to influenza infection of MLT-deficient mice with the effects of exogenous MLT administered at very high doses and by extremely different treatments. Table 1 shows the accessible preclinical studies investigating the possible therapeutic effects of MLT against viral diseases.

It seems noteworthy to make the observation that early studies used supraphysiological doses of MLT administered according to a circadian schedule, while the recent ones employed high pharmacological doses and typically did not follow any circadian administration. Probably, this divergence reflects a different conceptual approach connected to the MLT property to be exploited in fighting the infection. In the initial studies, the authors investigated whether the immunoenhancing action could be used to fight the disease, while in the latest ones, the authors mostly focused on the antioxidant and anti-inflammatory effects.

## 6. MLT and Bacterial Infections

The first evidence suggesting that MLT could influence the outcome of bacterial infection was its protective effect in an animal model of septic shock. A single injection of MLT, a few hours after intraperitoneal inoculation of a lethal dose of LPS in mice, was able to protect the animals. The doses used were 1, 2, 3, 4, 5 and 10 mg/kg b.w., and the protective effect, which involved a reduction in NO synthesis, was significant in the 2–5 mg range but lost at 10 mg [79]. This finding was then confirmed and extended in a variety of animal models and in humans with sepsis [80]. In particular, MLT could ameliorate the clinical status and increase the survival of human newborns with sepsis [79]. Doses and treatment schedules ranged from oral administration of two single doses of 10 mg within 12 h of the diagnosis of sepsis to one injection of 20 mg/kg in septic newborns treated with antibiotics. In general, the effect of MLT is suggested to depend on the suppression of pro-oxidant and pro-inflammatory pathways [80,81]. However, a recent study shows that polymicrobial sepsis in mice enhanced the expression of MT2 receptors in neutrophils and MLT administration protected the mice by enhancing the bactericidal effects of neutrophils [82]. In this study, MLT was used at massive doses of 50 mg/kg in vivo and 100 μg/mL in vitro [82]. Another study reports that in mice exposed to short photoperiods and infected with *Staphylococcus aureus* or *Escherichia coli*, MLT administration at 10 mg/kg resulted in the improved clearance of bacteria from blood [83]. A further in vitro study using porcine macrophages claims that the impracticable concentration of 1 mM of MLT may improve the bacterial clearance of enterotoxigenic *Escherichia coli* and suggests that MLT is important for controlling this type of infection [84]. Similarly, in a model of *Escherichia coli* meningitis, mice were treated for 7 days with MLT at 10, 30 and 60 mg/kg, and the claim was that MLT may prevent meningitis by acting on the intestinal microbiota [85]. Also, bacterial mastitis and infection with *Klebsiella pneumoniae* are among the bacterial diseases in which MLT is suggested to exert a therapeutic action by virtue of its anti-inflammatory and antioxidant effects [86,87]. The doses and concentrations of MLT used in these studies are in line with those in the above-reported citations.

Antimicrobial resistance is a growing emergency in public health. In particular, the transferable resistance–nodulation–division efflux pump TMexCD1-TOprJ1, conferring resistance to tigecycline, is becoming a serious health problem. A potentially very interesting and novel approach for combatting resistance to tigecycline used MLT, either in vitro or in vivo, in an infection model using *tmexCD1-toprJ1*-positive *Klepsiella pneumoniae* with encouraging results [88]. However, even in this study, MLT was used at extremely high concentrations (2–8 mg/mL) and doses (50 mg/kg) [88]. Table 2 lists the preclinical in vivo studies of the effects of MLT on bacterial infections.

As for viral diseases, also in these studies, the rationale for using gigantic amounts of MLT is not mentioned. It is somewhat surprising that in most studies, the well-known idea that circadian rhythms influence the outcome of and the susceptibility to infections [89] was completely ignored. In addition, both viral and bacterial infections may disrupt the circadian machinery [89], but whether such effects involve the immune–pineal axis or are exerted only on peripheral circadian clocks is still obscure. In my opinion, this is probably the crucial point that has to be carefully pondered in future studies aimed at improving the therapeutic approach for the use of MLT in infectious diseases. 

## 7. MLT and Parasitic Infections

The available evidence suggests that MLT may affect parasitic infections by acting directly on the biology of protozoan parasites and/or the host’s immune response. 

### 7.1. Malaria

Malaria is caused by parasites of the genus *Plasmodium*. The infection is transmitted by female *Anophele* mosquitos, which inject the parasite sporozoites into the host during blood feeding. The sporozoites establish the primary infection in the liver, where they proliferate and become merozoites. These are then released from the hepatocytes and infect erythrocytes, where they develop through a series of different stages and, finally, are released to infect more erythrocytes. The periodic rupture of erythrocytes releases cytosolic substances and parasite metabolites that elicit the host response, causing malaria symptoms, which follow a circadian cycle. In this rhythm, MLT plays a major role in synchronizing the *Plasmodium* cell cycle by acting on the cAMP-PKA and IP3-Ca^2+^ pathways to favor the synchronous egress of the merozoites, which enhances their capacity to invade circulating erythrocytes. The development of MLT-related analogues capable of disrupting this cycle shows a promising therapeutic potential against malaria, which affects several million people worldwide [90]. Remarkably, a recent report claims that MLT administered at doses of 5 and 10 mg/kg may prevent brain damage and cognitive impairment in an animal model of cerebral malaria [91]. In this model, mice were infected with *Plasmodium berghei*, which develops an asynchronous pattern of infection because its life cycle is not influenced by MLT [92]. 

### 7.2. Trypanosomiasis

*T. cruzi* is transmitted by an insect vector or via blood transfusion and organ transplantation. In humans, it causes Chagas’ disease, an extremely debilitating illness that has spread through migration from Latin America to the rest of the world, especially to the United States and Europe [93]. 

The possible therapeutic effect of MLT against *T. cruzi* have been studied in rodents.

In a series of studies by Santello and coworkers, the administration of MLT at 5 mg/kg either before infection as a pretreatment or during *T. cruzi* infection in rats proved to protect the animals against the disease by increasing their Th1 response while suppressing the Th2 response, as evidenced by the enhanced production of TNF-α, IFN-γ, IL-12 and IL-2 and increased leukocyte counts [94,95,96].

In the same model, another study showed that MLT and zinc treatment increased the plasma levels of IL-2 and IL-10, as well as thymocyte proliferation, counteracting the parasite-induced immune alterations [97]. A more recent study using mice reports that MLT decreased the circulating load of parasitemia without affecting parasite replication. At the cellular level, MLT notably enhanced parasite release, a potentially dangerous effect [98]. 

Human African trypanosomiasis is caused by the protozoan parasites *Trypanosoma brucei*, and it is transmitted by the bite of the tsetse fly. The disease is also called sleeping sickness because it is associated with the disruption of the circadian rhythms, possibly because it elicits a Th1-skewed immune response [99]. However, apart from a publication showing that MLT administration restored a normal sleep pattern in rats infected with *T. brucei* [100], no study has defined a possible therapeutic action of MLT against the infection.

### 7.3. Leishmaniases

Leishmaniases are caused by protozoan parasites of the *Leishmania* genus transmitted by the bite of a sand-fly and characterized by cutaneous, mucocutaneous or visceral lesions. *Leishmania* amastigotes are obligatory intracellular parasites that can live and reproduce within macrophages. By doing so, the parasite is able to modulate the host immunity, reducing inflammation and the adaptive immune response [101].

The first evidence of a possible MLT action in Leishmaniases was provided in 2014 by an in vitro study showing that a pharmacological concentration of MLT could reduce the number of viable *Leishmania infantum* promastigotes by altering Ca^++^ distribution in the parasite [102]. An interesting study in rodents shows that the Leishmania/host interaction varies following the circadian rhythm of MLT production and that MLT treatment during day time reduced the macrophage uptake of arginine by 40%, inhibiting parasite replication [103]. More recently, a report shows that pharmacological concentrations of MLT reduced *Leishmania amazonensis* infection in murine macrophages and modulated host microRNA expression, as well as the production of cytokines such as IL-6, MCP-1/CCL2 and RANTES/CCL9 [104]. A remarkable study combined MLT with amphotericin B in solid lipid nanoparticles and treated *Leishmania donovani*-infected BALB/c mice, inducing a 98.89% decrease in the intracellular parasite load in liver tissue. It was, therefore, emphasized that MLT would be effective in combating visceral Leishmaniases [105].

### 7.4. Toxoplasmosis

Toxoplasmosis is caused by the obligate intracellular parasite *Toxoplasma gondii* and is one of the most common parasitic infection in humans. Toxoplasmosis is very frequent and usually asymptomatic, but it may cause a fatal disease in presence of immunodeficiency. In spite of this risk, the therapeutic options against severe Toxoplasmosis are still inadequate [106].

As far as it concerns MLT, few studies addressed its possible role against Toxoplasmosis. In a model of retinochoroiditis, rats were infected with *Toxoplasma gondii* and treated with MLT (3mg/kg) for one month. The experimental groups also included pinealectomized mice and zinc supplementation. The highest cellular infiltration by CD3+, CD4+ and CD8+ cells was observed in the choroids and retinas of rats treated with MLT in combination with zinc supplementation. On the contrary, the minor infiltration was found in pinealectomized animals not supplemented with zinc [107]. Similar findings were obtained in other animal studies conducted by the same group [108,109].

Although at concentrations in the mM range, MLT has been found to inhibit *Toxoplasma gondii* growth in a monkey kidney cell line culture without affecting host cells’ viability. The authors suggested that MLT reduced parasite growth, inducing both apoptosis and necrosis by modifying energy metabolism [110]. An opposite finding was reported in a human colon adenocarcinoma cell line, where 0.2 nM of MLT was proved to boost parasite replication. It was also suggested that *Toxoplasma* parasites may degrade IDO1 and convert tryptophan to MLT, which, in turn, suppress ROS generation and favors their growth. The presence of IFNγ prevents IDO1 degradation so that tryptophan is catabolized into kynurenine, inducing cell death [111].

The impractical high concentrations of MLT used in these in vitro studies and the conflicting results obtained cast doubts on the biological meaning of these results.

Curiously, in most in vivo studies related to parasitic infections, MLT was administered at pharmacological doses (few mg/kg), without, however, reaching the exaggerated doses used in the models of viral and bacterial infection. Perhaps, this peculiarity reflects the more specific and higher sensitivity of parasites to MLT in comparison to pathogenic viruses and bacteria.

## 8. Conclusions

There is no doubt that being a multitasking molecule, MLT may influence infectious diseases. In this sense, the more persuasive studies are those showing that MLT-deficient animals have higher susceptibility and/or higher mortality when infected with influenza virus H1N1 [78], *Leishmania amazonensis* [103] and *Toxoplasma gondii* [107]. Other outstanding studies are those showing that MLT may overcome tigecycline resistance in a model of *Klebsiella pneumoniae* infection [88], as well as its synergy with amphotericin B in combating Visceral Leishmaniases [105]. These observations suggest that combining MLT with antiviral drugs or antibiotics might be a very promising approach for fighting infections. In fact, due to its equilibrating action on the immune response and its anti-inflammatory effect, MLT might improve the therapeutic index of the drugs and decrease their toxicity. On the other hand, in oncology, MLT has already been shown to improve the effectiveness of anti-cancer treatments while counteracting their negative side effects [112]. The majority of the experimental studies agree that MLT exerts positive therapeutic effects in viral, bacterial and parasitic infections. This evidence encouraged the performance of clinical studies in sepsis patients and COVID-19 patients. The results obtained confirmed the therapeutic potential of MLT against sepsis [80,81] but were controversial in the case of COVID-19 [67,68,69,70,71,72,73,74,75]. In general, the beneficial action of MLT was ascribed to its immunomodulating and/or anti-inflammatory properties. Nevertheless, no investigation challenged the hypothesis that these properties might be two sides of the same coin, i.e., part of the immune–pineal axis, whose physiological role is balancing innate and adaptive immune responses for the sake of optimizing the defense against the invading pathogen. In fact, in all relevant in vivo studies, MLT has been administered using a variety of treatment schedules and very different doses, mostly in the high pharmacological range. Similarly, MLT was often tested in vitro for possible direct effects on infectious agents at very high, almost unreasonable concentrations. Moreover, no study asked whether the effects observed were due, at least in part, to the ability of MLT to synchronize the circadian machinery and, hence, all the neuroendocrine, immune and hematopoietic cycles influencing either the susceptibility or the resistance to infectious agents. Thus, the state of the art in this potentially important topic is far from being clear and does not provide a sound basis for clinical applications. It is in fact very possible that to be most effective, MLT has to be administered at different doses and different schedules according to the infection stage. It might be true that in the early stages, when the host reaction is mostly of the innate type, MLT should be administered at pharmacological doses without taking into consideration any circadian rhythmicity, while later on, when adaptive immunity takes place, the MLT doses should be lower and administered in the evening in accordance with its endogenous circadian rhythm. This schedule is schematized in Figure 2. 

On the basis of the available evidence of the possible therapeutic actions of MLT in combating infections, various treatments schedules have been proposed. The scheme represents different experimental possibilities for investigating the therapeutic effects of MLT. The effects of pretreatment and treatment might be evaluated either separately or in combination, and within the treatment protocol, we could evaluate separately the effects of the non-circadian high doses vs. those of the circadian low doses and so on. Hopefully, various doses of MLT will be investigated, with low doses possibly ranging from 10 μg to 1000 μg/kg administered in a circadian fashion in the evening and high doses ranging from 3 mg to 50 mg/kg administered in the morning. Of course, all relevant control groups should be included.

In this way, the treatment schedule would mimic the physiological response of the immune–pineal axis, reinforcing both the anti-inflammatory action early during the infection and, later, the ensuing adaptive immune response. To verify this hypothesis further, studies investigating doses and treatment schedules in the various types of infection are clearly needed.

## Figures and Tables

**Figure 1 biomolecules-14-00356-f001:**
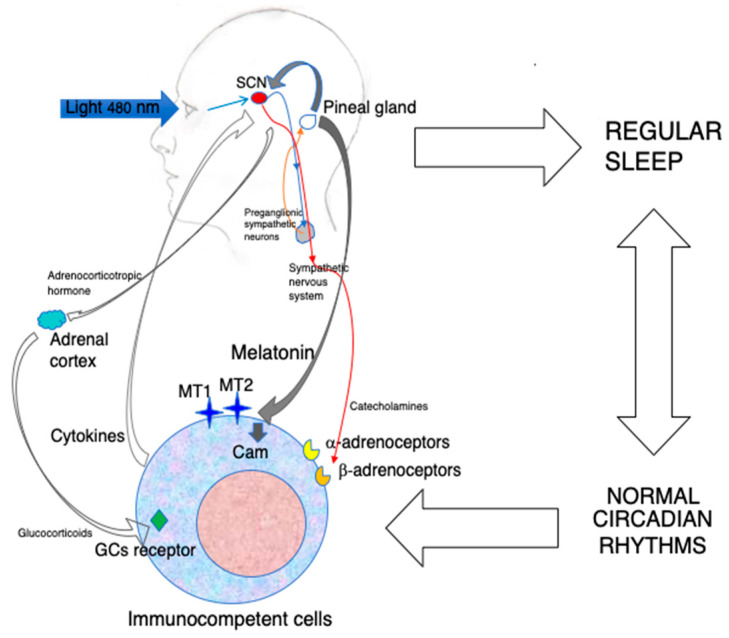
The indirect and direct immunoenhancing action of MLT.

**Figure 2 biomolecules-14-00356-f002:**
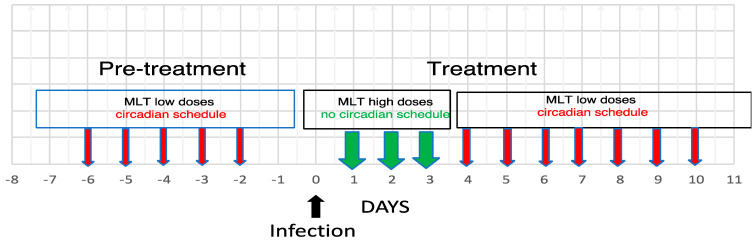
A possible experimental approach integrating the anti-inflammatory and immunoenhancing actions of MLT against infections.

**Table 1 biomolecules-14-00356-t001:** In vivo preclinical studies of the antiviral activity of MLT.

Pathogen	Species	MLT Dose	Treatment	Outcome	Ref.
EMCV	Mice	1 μg/mouse	10 days at 4 pm	Reversal of stress-induced death	[57]
SFV	Mice	500 μg/kg	From 3 days before until 10 days after infection at 4 pm	Increased survival and decreased viremia	[58]
aWNV	Mice	5 μg/mouse	From 2 days before until 8 days after infection at 4 pm	Reduced mortality	[58]
VEEV	MIce	1 mg/kg	From 3 days before until 10 days after infection at 6 pm	Increased survival, decreased viremia and increased antibody response	[59]
RSV	Mice	5 mg/kg	Twice daily for 3 days	Reduced oxidative damage of the lung	[63]
RHDV	Rabbits	20 mg/kg	0, 12 and 24 h after infection	Decreased mitophagy, inflammation and innate immunity	[64,65]
H1N1	Mice	3, 10, 30 mg/kg	Pretreatment for 3 days before infection	Decreased lung injury by the inhibition of mast cells and cytokine storm	[78]
H1N1	Mice	200 mg/kg	6 h before and 2, 4 and 6 days post-infection	Inhibition of pro-inflammatory cytokines and stimulation of IL-10; synergy with an antiviral drug	[76]
H3N2	Mice	30 mg/kg	For 7 days at 6 pm	Attenuated pulmonary damage, leukocyte infiltration and edema	[77]

The features of the existing preclinical studies of the possible therapeutic effects of MLT in viral infections are reported. EMCV: encephalomyocarditis virus; SFV: Semliki Forest virus; aWNV: attenuated West Nile virus; VEEV: Venezuelan equine encephalitis virus; RSV: respiratory syncitial virus; RHDV: rabbit hemorrhagic disease virus; H1N1, influenza A H1N1; H3N2: influenza A H3N2.

**Table 2 biomolecules-14-00356-t002:** Preclinical and clinical studies on the anti-bacterial effects of MLT.

Pathogen	Species	MLT dose	Treatment	Outcome	Ref.
Lethal dose of LPS	Mice	1, 2, 3, 4, 5 and 10 mg/kg	3 or 6 h after LPS injection	2, 3, 4 and 5 mg/kg reduced mortality and NO synthesis	[79]
Sepsis	Human Newborns	2 × 10 mg	Oral administration within 12 h after diagnosis	Increased survival and improved clinical status	[80]
Sepsis	Human newborns	20 mg/kg	One injection plus antibiotics	Increased survival and improved clinical status	[81]
Polymicrobial sepsis	Mice	50 mg/kg	Two doses, 30 min before and 30 min after cecal ligation puncture	Protection of mice via the induction of neutrophil extracellular trap	[82]
*Staphylococcus aureus* and *Escherichia coli*	Mice	10 mg/kg	Once daily for 7 days	Improved clearance of bacteria from blood, reduced iNOS, plasma C-reactive protein and COX2 expression in the hypothalamus.	[83]
*Escherichia coli*	Mice	30 mg/kg	Pretreatment for 7 consecutive days before infection	Prevention of and protection from bacterial meningitis by modulating the intestinal microbiota	[85]
Tigecyclin resistant *Klebsiella pneumoniae*	Mice	50 mg/kg	One dose after infection	Restoring tigecycline activity	[88]

The available preclinical and clinical studies of the therapeutic effects of MLT against bacterial infections are reported. NO: nitric oxide; iNOS: inducible nitric oxide synthetase; COX2: cyclooxygenase

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
