# Peer review of "Role of Melatonin in Viral, Bacterial and Parasitic Infections"

_biomolecules, 2024, doi:10.3390/biom14030356_

Round 1
Reviewer 1 Report
Comments and Suggestions for Authors
In the review article titled “Role of melatonin in viral, bacterial and parasitic infections” Maestroni carries out a careful and exhaustive review of the scientific literature regarding the role of melatonin in infectious diseases and in the interaction between host and infecting agents including viruses, bacteria, protozoa, etc.
The review is very complete, comprehensively addressing the many issues inherent to methodological differences and treatment schemes in animal models and humans.
The literature reviewed is extensive and up-to-dated.
MINOR CONCERN
It is advisable to improve the graphic characteristics of the figure, in order to make it adequate for the good level of compilation of the review
There are few typos. I recommend accurate proofreading of the manuscript and I suggest corrections
A few examples:
IL-1b, TNF-a, IFNg, etc: please use Greek symbols
159 MLT may inhibit NK-kB… please correct with NF-kB
160Some of these MLT effects seems… please correct with seem
278 did not follow any circadian radmin-… please correct with administration
Comments on the Quality of English LanguageMinor corrections suggested
Author Response
I wish to thank this reviewer for his comments.
All the indicated typos have been amended and the manuscript has been carefully read and corrected. The corrections show in red.
Figure 1 has been modified and hopefully improved
Reviewer 2 Report
Comments and Suggestions for Authors
The author wrote a nice review article about the role of melatonin and viral, bacterial and parasitic infections. The references are both relevant and recent, and the cited sources are correctly referenced. I enjoyed reading the article and its review about the topic. However, I do detect some mistakes and inconsistencies that need to be addressed.
-Melatonin synthesis in animals involves serotonin N-acetyltransferase (NAT) and N-acetylserotonin O-methyltransferase (ASMT; formerly known as hydroxyindole-O-methyltransferase, HIOMT). As far as I know COMT is involved in plants, so I suggest correcting this. (lines 65-66)
- The author wrote in Lines 36-37 “In general, the concentration of extra-pineal MLT is reported to be order of magnitude higher than that of the circulating pool derived from the pineal gland” and then, in lines 79-81 “the concentration of extra-pineal MLT is usually three orders of magnitude higher than that of plasma MLT”. Please clarify the order of magnitude you are referring to, if known.
- The information provided in Figure 1 on the immunoenhancing action of pineal melatonin is limited and lacks depth. A caption should be included to clarify the meaning of the colours of the arrows, abbreviations, and other symbols used.
- Table 1 is missing the species (mice) in which the study was carried out, as cited in reference 76.
- Anopheles (line 335), Staphylococcus aureus, Escherichia coli (both in table 2) should be written in italics.
- The headings 6.2 have been reprinted. Should be: 6.3. Leishmaniases, 6.4. Toxoplasmosis.
- The author suggests that there is a temporary suppression of pineal melatonin production and a stimulation of its synthesis in immune cells (macrophages/microglia) during an inflammatory process. A diagram or schematic would help the reader to understand.
Overall, the article appears to be well-written and comprehensible.
Author Response
Many thanks to this reviewer for his comments.
- The oversight concerning HIOMT has been corrected
- I used in both cases the term "orders".
- Figure 1 has been modified.
- Table 1 has been corrected
- The font in table 2 has been corrected
- The heading of 6.2 has been corrected
- The existence of the immune-pineal axis is not a suggestion of mine, there are several detailed reports about it ( https://pubmed.ncbi.nlm.nih.gov/?term=The+immune-pineal+axis). In any case, I added a figure showing a possible experimental treatment schedule integrating both the anti-inflammatory and immunenhancing MLT actions that would mimic the physiological response of the immune-pineal axis.
Round 2
Reviewer 2 Report
Comments and Suggestions for Authors
The author has addressed some of the reviewers' requests. However, some minor issues remain to be addressed.
-All scientific names of species should be in italics as they are written in Latin. Please review them.
-The numbering in Section 6 is still wrong.
Author Response
I amended the manuscript according to the reviewer comments.